# Direct Visualization of UV-Light on Polymer Composite Films Consisting of Light Emitting Organic Micro Rods and Polydimethylsiloxane

**DOI:** 10.3390/polym14091846

**Published:** 2022-04-30

**Authors:** Misuk Kim, Jiyoun Kim, Hyeonwoo Kim, Incheol Jung, Hojae Kwak, Gil Sun Lee, Young Jun Na, Young Ki Hong, Dong Hyuk Park, Kyu-Tae Lee

**Affiliations:** 1Department of Physics, Inha University, Incheon 22212, Korea; kim.ms@inha.ac.kr (M.K.); nkim5778@naver.com (H.K.); jungin1107@gmail.com (I.J.); khj0124@inha.edu (H.K.); 2Department of Chemical Engineering, Program in Biomedical Science and Engineering, Inha University, Incheon 22212, Korea; freejiyoun@gmail.com; 3Department of General Education, Kookmin University, Seoul 02707, Korea; gslee@kookmin.ac.kr; 4Department of Physics, Gyeongsang National University, Jinju 52828, Korea; oejson@gmail.com; 5Research Institute of Natural Science, Gyeongsang National University, Jinju 52828, Korea

**Keywords:** ultraviolet, Alq3, micro-rods, PDMS, polymer composite films, visualization

## Abstract

We experimentally demonstrate the direct visualization of ultraviolet (UV) light using flexible polymer composite films consisting of crystalline organic tris-(8-hydroxyquinoline) aluminum (Alq3) micro-rods and polydimethylsiloxane (PDMS). The representative organic mono-molecule Alq3, which is a core material of organic light-emitting diodes, was used to detect light in the invisible UV region and visualize photoluminescence (PL). Alq3 shows absorption in the UV region and light-emitting characteristics in the green region, making it an optimal material for UV visualization because of its large Stokes transition. Crystalline Alq3 micro-rods were fabricated in a deionized water solution through a sequential process of reprecipitation and self-assembly. Highly bright photoluminescence was observed on the highly crystalline Alq3 micro-rods under UV light excitation, indicating that the crystalline structures of Alq3 molecules affect the visible emission decay of excitons. The Alq3 micro-rods were manufactured as flexible polymer composite films using a PDMS solution to observe UV photodetector characteristics according to UV intensity, and it was confirmed that the intensity of the fine UV light reaching the earth’s surface can be visualized by making use of this UV photodetector.

## 1. Introduction

Ultraviolet (UV) rays are electromagnetic waves present in the sunlight. Because of their relatively high energy, they are invisible to the naked eye and are harmful when they reach our eyes from the sun. Many researchers have made great efforts to detect UV rays and have mainly conducted research using inorganic-based semiconductors or their hybrid nano-architectures [1,2,3,4]. Through the detection of ultraviolet rays, applications such as imaging, medical sensing, secure communication, and measurement of various surrounding environments are possible, and active research is in progress [5,6,7,8]. A major issue in UV light detection is its effective operation owing to its increased sensitivity, stability, and efficiency [9,10,11,12]. However, the UV detection devices are coupled with electrical devices through high-cost complicated manufacturing processes, and visualization is required to recognize UV exposure immediately. One of the effective simplified sensors, a colorimetric sensor that detects UV light by a chemical reaction of a polymer when exposed to UV light, is difficult to capsule because it is manufactured in a solution state [13]. In the case of UV sensors using diacetylene, only studies on the reactivity of diacetylene according to the UV exposure time have been reported, and studies on detecting indicators according to the power of UV light are insufficient [14,15].

Among optically active organic materials, tris-(8-hydroxyquinoline) aluminum (Alq3) is used as a light-emitting layer in light-emitting diodes and is widely studied as an active layer in solar cells because it absorbs ultraviolet light owing to the Stokes shift effect and produces light in visible light areas [16,17,18,19,20]. In addition, Alq3 is a suitable indicator for UV photodetectors because it emits light in the green region (~550 nm), which the eyes accept most comfortably and sensitively [21]. These features enable the visualization of harmful UV light with powerless indicators and sensors through the confirmation of environmental information by absorbed light [22,23].

In this study, we manufactured a solid-state composite film by mixing a polymer, polydimethylsiloxane (PDMS), which is easier to manufacture than the capsule design, and its reactivity varies depending on the power of UV light, in both the polymer composite film and solution states. In addition, UV light was visualized in various ways, such as printing the Alq3 micro-rod solution directly into the clothes in the desired shape and confirming that the print on the clothes reacts to UV light.

## 2. Materials and Methods

### 2.1. Experimentals

Alq3 (C_27_H_18_AlN_3_O_3_, purity 99.995%) and sodium dodecyl sulfate (SDS, CH_3_(CH_2_)_11_OSO_3_Na, purity 99.0%) were purchased from Sigma-Aldrich (St. Louis, MO, USA). Alq3 was dissolved in tetrahydrofuran to a concentration of 3 mg mL^−1^. The Alq3 solution was injected into the vigorously magnetically stirred solution, which was dissolved in deionized (DI) water and SDS (1 mg·mL^−1^), [24] and then heated to 70 °C for 1h and homogenized using a magnetic stirrer for complete dissolution. For the hybrid film using the Alq3 solution and PDMS, 20 *v*/*v*% of the Alq3 solution grown by reprecipitation was added to the SYLGARDTM 184 Silicone Elastomer Kit purchased from DOW. PDMS, curing agent, and Alq3 micro-rod solution were mixed in a ratio of 10:1:2. The Alq3/PDMS mixture was stored at room temperature in a low vacuum state (approximately 0.03 MPa) for 6 h and then dried again in a vacuum oven at 45 °C for 6 h, as shown in Figure 1.

### 2.2. Measurements

To confirm the formation of the micro-rods, we used field-emission scanning electron microscope (FE-SEM; Hitachi, Tokyo, Japan, SU-8010) at an accelerated voltage of 15 kV. Portable UV lamp was utilized to irradiate light with a wavelength of ~365 nm on the micro-rods to observe the light-emitting variation of the Alq3 micro-rods according to the magnitude of the UV light power. The distance between the light source and sample was adjusted, and images were taken every 50 μW with increasing UV light power.

The photoluminescent (PL) spectra of the micro-rods were measured using a customized laser confocal microscope (LCM) [25]. The 405 nm line of an unpolarized diode laser was used for excitation of the LCM PL. The organic materials were placed on a glass substrate mounted on the xy-stage of the confocal microscope. An oil-immersion objective lens (NA of 1.4) was used to focus the unpolarized laser light on the crystal. The spot size of the laser beam focused on the sample was approximately 200 nm. The scattered light was collected using the same objective lens, and the excitation laser light was filtered using a long-pass edge filter (Semrock, Rochester, NY, USA). The red-shifted PL signal was focused onto a multimode fiber (core size = 100 µm), which acted as a pinhole for confocal detection.

The other end of the multimode fiber was connected to a photomultiplier tube to acquire the PL image, or the input slit of a 0.3 m long monochromator equipped with a cooled charge-coupled device to acquire the PL spectra. Solid-state PL values were measured on the nanometer scale. The laser power incident on the sample and the acquisition time for each LCM PL spectrum were fixed at 500 µW and 1 s, respectively, for all the confocal PL measurements.

## 3. Results and Discussion

### 3.1. Alq3 Miroc-Rods

The crystalline one-dimensional Alq3 micro-rods were grown by the self-assembly method in DI water with the help of surfactants, and this was confirmed the Alq3 micro- rods through the FE-SEM image, as shown in Figure 2a,b. The micro-rods are approximately 10–20 µm in length and 500 nm–2 µm in diameter, indicating homogeneity. The cross-section of the micro-rods was a hexagon in Figure 2c. We confirmed that the hexagonal Alq3 micro-rods with the crystallinity were luminescent and brighter than the amorphous Alq3 micro-rods from the previous experiments [24]. These crystal Alq3 micro-rods are more responsive to the UV light. In an experiment conducted to verify the absorption spectrum of Alq3, the Alq3 solution absorbed UV light at 380 nm, resulting in a PL peak at 530 nm, as shown in Figure 2d. This is a Stokes shift phenomenon, in which the energy of the emitted photon becomes less than that of the absorbed photon. The Stokes shift of this Alq3 material is approximately 150 nm (~0.923 eV), which is larger than that of a general light-emitting material. From this result, we can conclude that Alq3 with a relatively large Stokes shift is a potential sensor that allows direct visualization of UV light by the naked eye and visually displays the luminous image.

### 3.2. Direct Visualization of UV Light Using Alq3 Miro-Rod and PDMS Composite Films

To visualize UV light directly with the naked eye without the aid of any device, we fabricated a polymer composite film comprising crystalline Alq3 micro-rods and PDMS, as shown in Figure 1. We conducted an experiment on the luminescence characteristics according to the intensity of UV light. For an accurate reactivity measurement of UV detection using the produced polymer composite film, the power reaching the sample was measured, and the optical properties of the Alq3/PDMS polymer composite film at 10, 20, 50, and 100 µW were measured according to the intensity.

The Alq3/PDMS film in Figure 3a, when UV light is not exposed, appears translucent and light yellow. However, when UV lights are irradiated on the polymer composite sample in Figure 3b–f, the Alq3/PDMS polymer composite film shows the vivid green and brighter light emissions proportional to the magnitude of the UV lamp power. Consequently, as the exposure intensity of ultraviolet light reaching the sample surface increased, the Alq3/PDMS polymer composite film became increasingly luminous. At the highest power of 100 µW, the Alq3/PDMS film could be seen glowing like white light. The color and brightness of the light-emitting Alq3/PDMS polymer composite film varies depending on the intensity of the light being irradiated. The weakest intensity of UV light reaching the Earth’s surface from natural sunlight is of the order of a few microwatts. UV rays of this intensity are known to have a detrimental effect on human skin, and our sample confirmed a green luminescence visible to the naked eye even at a light intensity of approximately 5 µW. For a more precise analysis of the luminescence characteristics of the Alq3/PDMS polymer composite film, we measured the PL spectra using the LCM PL equipment, as shown in Figure 3g. In addition, the power of the UV lamp was controlled at 5, 10, 20, 50, and 100 µW for the Alq3/PDMS polymer composite film. The main PL peak of the Alq3/PDMS polymer composite film was observed at 520 nm, similar to the unique PL peak of the crystalline Alq3 micro-rods. The magnitude of the irradiated UV light changed, but only the PL intensities differed. As the power of the UV lamps was increased, the PL intensity of the Alq3/PDMS polymer composite film increased. Through nanoscale LCM PL experiments for the Alq3/PDMS polymer composite film, the intrinsic PL spectral characteristics of Alq3 were confirmed even under weak light of 5 µW, which confirms our polymer composite film to be a very sensitive UV visualization sample for the naked eye.

### 3.3. Demonstration of Direct UV Light Visuzlization Using Crystalline Alq3 Micro-Rods Printed T-Shirt Clothing

In order to check whether visualization is possible when the light-emitting Alq3 micro-rods are attached to various materials with complex shapes, we performed an experiment to check whether UV visualization is possible by printing on clothes, and the visible and immediate reactions to UV light confirmed the possibility of printing crystalline Alq3 micro-rod solutions on clothes. Figure 4a illustrates the experimental visualization concepts using the crystalline Alq3 micro-rod solution. We made a simple plastic mask of the desired shape. The crystalline Alq3 micro-rod solution was sprayed on clothes at a distance of 30 cm or more, similar way like a stencil method. After spraying on the cloth, the printed patterns were clearly visible under the UV light lamp. Figure 4b,c compared the digital camera photo images taken before and after the UV light of the 20 μW power was irradiated. After the UV lamps were turned on, the invisible printed patterns on the cloth were clearly visible.

## 4. Conclusions

For direct visualization of UV light by the naked eye, we fabricated crystalline Alq3 micro-rods and observed their luminescence characteristics. The Alq3 micro rods, which were approximately 10 µm length and 500 nm–2 μm in diameter and had a hexagonal cross-section, absorbed an incident light of wavelength 380 nm and exhibited a large Stokes shift with a PL peak at 530 nm. To develop the UV photodetector, a polymer composite film was fabricated using PDMS, and its potential as a UV sensor was confirmed. The Alq3/PDMS polymer composite film changed to a clear bright green light under incident UV light. The intensity of PL, measured using LCM PL equipment, increased with increasing intensity of UV lamps, which confirmed the light-emitting characteristics of the Alq3/PDMS film. Because of visible green light emission, even under the stimulation of very weak UV rays, the Alq3/PDMS polymer composite film was confirmed to be suitable as a very sensitive UV visualization sample for the naked eye. Finally, we confirmed that UV light could be detected on the T-shirt by spraying crystalline Alq3 micro-rods solution.

## Figures and Tables

**Figure 1 polymers-14-01846-f001:**
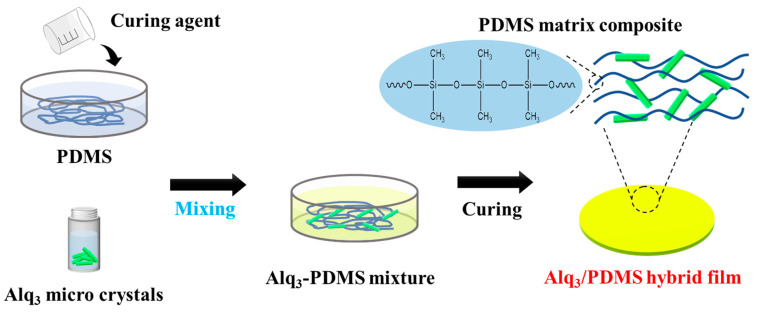
Schematic diagram of Alq3/PDMS hybrid film fabrication process.

**Figure 2 polymers-14-01846-f002:**
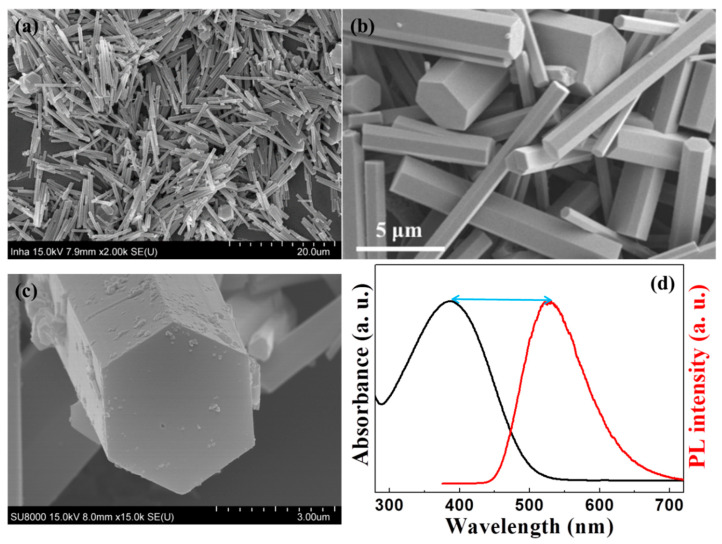
(**a**,**b**) SEM images of Alq3 microcrystals. (**c**) Magnification side-view of SEM image of hexagonal-shape Alq3 micro-rod. (**d**) Comparison of the absorption with its PL of the Alq3 microcrystals in the solution, showing a relatively large stock shift.

**Figure 3 polymers-14-01846-f003:**
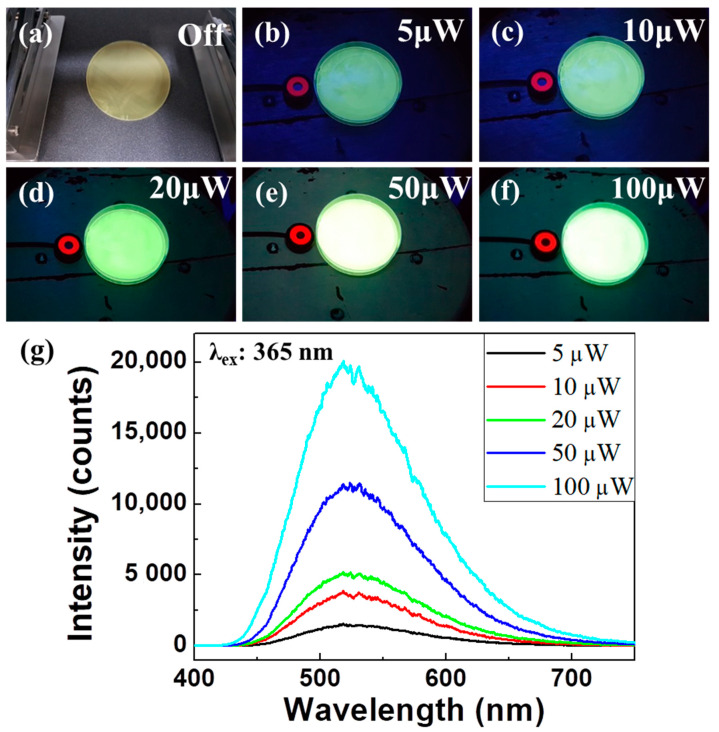
(**a**–**f**) Digital camera images of Alq3/PDMS polymer composite films, when irradiated with UV lamp power depending on the power of (**a**) 0 μW, (**b**) 5 μW, (**c**) 10 μW, (**d**) 20 μW, (**e**) 50 μW, and (**f**) 100 μW, respectively. (**g**) LCM PL results of the corresponding to the previous luminescence experimental image depending on the exposure power.

**Figure 4 polymers-14-01846-f004:**
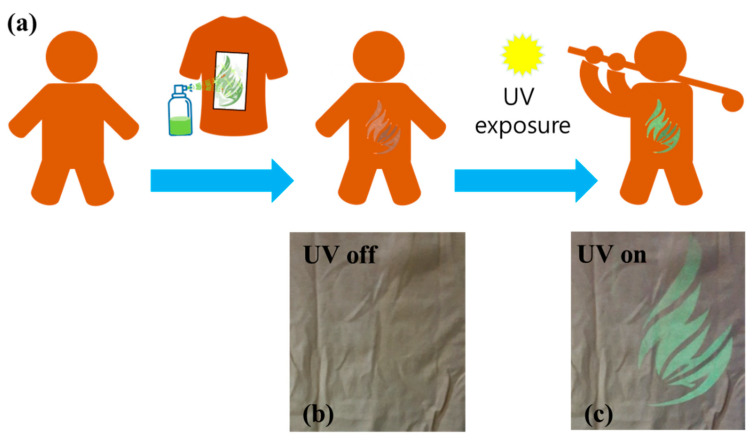
(**a**) Schematic illustrations of UV visualizations on the T-shirt by spray-printing the crystalline Alq3 micro-rod solution. (**b**,**c**) Magnification of the Alq3 solution printed parts in the T-shirt. UV lamp (maximum power of 20 μW) is turned off (**b**) and on (**c**).

## Data Availability

The data presented in the study are available on the request from the corresponding authors.

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
