# Peer review of "Direct Visualization of UV-Light on Polymer Composite Films Consisting of Light Emitting Organic Micro Rods and Polydimethylsiloxane"

_polymers, 2022, doi:10.3390/polym14091846_

Round 1
Reviewer 1 Report
In this paper, authors experimentally demonstrated the direct visualization of UV-Light using flexible polymer composite films consists of crystalline light emitting organic micro rods and polydimethylsiloxane. The presented results are interesting and significant. This manuscript is recommended to be published after minor revision. The suggested changes are as follows:
- As author shown SEM image of Alq3 microcrystals and magnification side-view of SEM image of hexag-onal-shape Alq3 micro rod. Authors should show the size distributions of organic micro rods.
- English should be improved to meet the requirement of publication.
- The Novelty of the manuscript is good.
- Why author used PDMS as a curing agent?
- Authors should add the measurements of typical I-V characteristics of the Alq3 LEDs.
In this paper, authors experimentally demonstrated the direct visualization of UV-Light using flexible polymer composite films consists of crystalline light emitting organic micro rods and polydimethylsiloxane. The presented results are interesting and significant. This manuscript is recommended to be published after minor revision. The suggested changes are as follows:
- As author shown SEM image of Alq3 microcrystals and magnification side-view of SEM image of hexag-onal-shape Alq3 micro rod. Authors should show the size distributions of organic micro rods.
- English should be improved to meet the requirement of publication.
- The Novelty of the manuscript is good.
- Why author used PDMS as a curing agent?
- Authors should add the measurements of typical I-V characteristics of the Alq3 LEDs.

Author Response
The authors thank the reviewers for their thorough reading of our manuscript and the useful comments. We have revised the manuscript based on the reviewers’ comments, which were highlighted in yellow color. We hope that the manuscript will be acceptable in the current form. The reviewers’ comments appear in black, and the authors’ responses in blue.

Reviewer 2 Report
The authors presented a concept of UV visualization by using a composite of a fluorescent emitter and a polymer. The motivation for designing such an experiment is really confused. Sunlight has always UV light, why does it have to be detected? Moreover, many of the fluorescent emitters exhibit the observation that absorbs UV light then emits fluorescence. There is not any scientific contribution in the manuscript. The other comment is that the authors did not describe why Alq3 nanorod is used because an amorphous ALq3 should have better photophysics characteristics.
Author Response

(The authors gave the same response as above.)

Reviewer 3 Report
To demonstrate the direct visualization of ultra-violet (UV) light in a more immediate and sustainable way, this paper devised a series of experiments by using polymer composite films consisting of polydimethylsiloxane (PDMS) and crystalline organic tris-(8-hydroxyquinoline) Alq3 micro rods, which display the absorption of the UV region and the light emitting characteristics of the green region. Structural characterization and some other properties were studied (SEC, absorbance, PL intensity and etc). This paper can be published although some revisions are required, and some specific comments are as follows:
- It is recommended to add somerecent references.
- In figure 3, it would be better to add up more samples under the powers of UV lamp such as 30and 40µW.
- In figure 3, intensity could be detected at 5µW, and it would be better to research the minimal UV light power that makesthe direct visualization possible.
- In figure 4, the information of the power of UV lamp was missed.
- The light emitting Alq3 micro rods can be attached to clothes made by different materials like cotton and wool so as to examine whether the visualization is possible.
Author Response

(The authors gave the same response as above.)

Round 2
Reviewer 2 Report
After considering the authors' comments, I still doubt the significance of applying UV visualization in sunlight. Even though the authors described a potential application to detect a slight UV light, the authors also can not able to provide the threshold power of detected UV light and a visualization standard. The simple images can not illustrate such an application is possible. Consequently, the manuscript is not accepted.